# Taqman PACMAN: a simple molecular approach for positive rapid antigen test confirmation during periods of low prevalence

Gregory R. McCracken,[1] Glenn Patriquin,[1,2] Todd F. Hatchette,[1,2,3,4] Ross J. Davidson,[1,2,3,4] Barbara Goodall,[4] Lisa Barrett,[1,2,3,4] James MacDonald,[1] Charles Heinstein,[1] Janice Pettipas,[5] John Ross,[6] Jason J. LeBlanc[1,2,3,4]

**ABSTRACT**    Antigen-based rapid diagnostic tests (Ag-RDTs) were widely deployed to enhance SARS-CoV-2 testing capacity during the COVID-19 pandemic. Consistent with national guidance for low prevalence settings, positive Ag-RDTs were confirmed using nucleic acid amplification tests (NAATs) to avoid false positive results. However, increasing demands for positive Ag-RDT confirmation competed with other testing priorities in clinical laboratories. This work hypothesized that real-time RT-PCR without nucleic acid extraction (NAE) would be sufficiently sensitive to support positive Ag-RDT confirmation. Ag-RDT and NAAT results from community-based asymptomatic testing sites prior to the omicron variant wave were compared to calculate the weekly false positive rate (FPR) and false detection rate (FDR). Real-time RT-PCR was compared with and without NAE using 752 specimens previously tested positive for SARS-CoV-2 using commercial NAATs and 344 specimens from Ag-RDT-positive individuals. The impact of SARS-CoV-2 prevalence on laboratory resources required to sustain Ag-RDT confirmation was modeled for the RT-PCR with and without NAE. Overall, FPR was low [0.07% (222/330,763)] in asymptomatic testing sites, but FDR was high [30.7% (222/724)]. When RT-PCR was compared with and without NAE, 100% concordance was obtained with NAAT-positive specimens, including those from Ag-RDT-positive individuals. NAE-free RT-PCR significantly reduced time to results, human resources, and overall costs. A 30.7% FDR reaffirms the need for NAAT-based confirmation of positive Ag-RDT results during low SARS-CoV-2 prevalence. NAE-free RT-PCR was shown to be a simple and cost-sparing NAAT-based solution for positive Ag-RDT confirmation, and its implementation supported data-driven broader Ag-RDT deployment into communities, workplaces, and households.

**IMPORTANCE**    Rapid antigen testing for SARS-CoV-2 was widely deployed during the COVID-19 pandemic. In settings of low prevalence, national guidance recommends that positive antigen test results be confirmed with molecular testing. Given the high testing burden on clinical laboratories during the COVID-19 pandemic, the high volume of positive antigen tests submitted for confirmatory testing posed challenges for laboratory workflow. This study demonstrated that a simple PCR method without prior nucleic acid purification is an accurate and cost-effective solution for positive rapid antigen test confirmation. Implementing this method allowed molecular confirmatory testing for positive antigen tests to be sustained as antigen testing was expanded into large populations such as workplaces, schools, and households.

**KEYWORDS**    SARS-CoV-2, COVID-19, rapid antigen, molecular, extraction, PCR, prevalence, false positive

Address correspondence to Jason J. LeBlanc, Jason.Leblanc@nshealth.ca.

The authors declare no conflict of interest.

D iagnostic testing for SARS-CoV-2 has been a key pillar of Canada's COVID-19 pandemic response. Early in the pandemic, SARS-CoV-2 detection relied almost

exclusively on nucleic acid amplification tests (NAATs) like reverse transcription polymerase chain reaction (RT-PCR) (1–5). Once antigen-based rapid diagnostic tests (Ag-RDTs) became licensed by Health Canada, they were rapidly deployed to enhance SARS-CoV-2 testing capacity (1, 6–11).

National guidance for the use of SARS-CoV-2 Ag-RDTs in Canada is consistent with the World Health Organization (WHO), where applications included community testing of individuals with symptoms compatible with COVID-19 or symptomatic screening of other individuals at high risk of COVID-19 (e.g., health care workers and close contacts of positive SARS-CoV-2 cases) (6, 7). These guidelines also recognized the value of Ag-RDTs for asymptomatic testing in vulnerable populations or as a public education and engagement tool, thereby increasing awareness and reinforcing public health measures to mitigate SARS-CoV-2 transmission (6). On 21 November 2020, Nova Scotia became the first Canadian province to deploy Ag-RDTs as a public engagement tool in asymptomatic community testing sites, with the added benefit of identifying SARS-CoV-2 in individuals not eligible for symptomatic testing (8–11). Ag-RDTs were later expanded for use in households, schools, and workplaces. Consistent with national guidance for low prevalence settings, confirmatory testing using NAATs was performed on all positive Ag-RDTs from their deployment and continued until community SARS-CoV-2 activity was widespread (6–11).

From a clinical laboratory perspective, Ag-RDT confirmation using NAAT added to the existing workload and competed with other testing priorities. To meet NAAT testing demands, hospital laboratories in the Canadian province of Nova Scotia used strategies like specimen pooling during periods of low SARS-CoV-2 activity, where specimens were combined and tested simultaneously (Fig. 1) (12–14). If a specimen pool result was SARS-CoV-2 negative, all specimens within the pool were considered negative. In contrast, if a specimen pool was SARS-CoV-2 positive, each pool member was subsequently tested individually to identify which specimen(s) contributed to the positive result. Given the high pre-test probability of a specimen being positive with NAAT from an individual testing positive with an Ag-RDT, there was a proportional increase in specimen pools requiring resolution as testing was expanded into workplaces, schools, and households. Alternative testing strategies were sought to streamline Ag-RDT confirmation without delaying the reporting of other priority specimens (e.g., hospitalized individuals or long-term care residents).

To confirm positive Ag-RDTs, this work evaluated the use of a real-time RT-PCR based on Taqman probe technology without prior nucleic acid extraction (NAE). While NAE-free RT-PCR protocols typically have a relatively lower sensitivity compared to the same RT-PCR paired with an NAE (15–17), this work hypothesized that NAE-free RT-PCR might be sufficiently sensitive to confirm positive Ag-RDTs. In our laboratory, the NAE-free RT-PCR was commonly referred to as the "Taqman PACMAN," describing the SARS-CoV-2 Taqman RT-PCR for Positive Antigen Confirmation with Molecular testing in Absence of a Nucleic acid extraction (PACMAN).

To demonstrate the value of the Taqman PACMAN, this study assessed the weekly Ag-RDT false positive rate (FPR; the proportion of individuals testing negative by NAAT who tested positive using Ag-RDT) and the weekly false discovery rate (FDR, the proportion of positive Ag-RDTs that did not confirm using NAATs). Both FPR and FDR have different merits and differ only in their denominator. The FPR represents how frequent false positives occur in patents without disease, which is dependent on the number of negative results identified. As such, the impact of false positive results could potentially be masked during periods of low prevalence with mass population testing of asymptomatic individuals, where the number of individuals testing negative is large. In contrast, the FDR is the number of false positive Ag-RDT results that occurred in the individuals testing Ag-RDT positive, therefore, in those suspected to have SARS-CoV-2. As these individuals were asked to self-isolate in this setting, a positive Ag-RDT might have caused individual impacts like financial and psychological stress. Compared to FPR, the FDR better reflects the proportions of individuals subjected to undue stress from

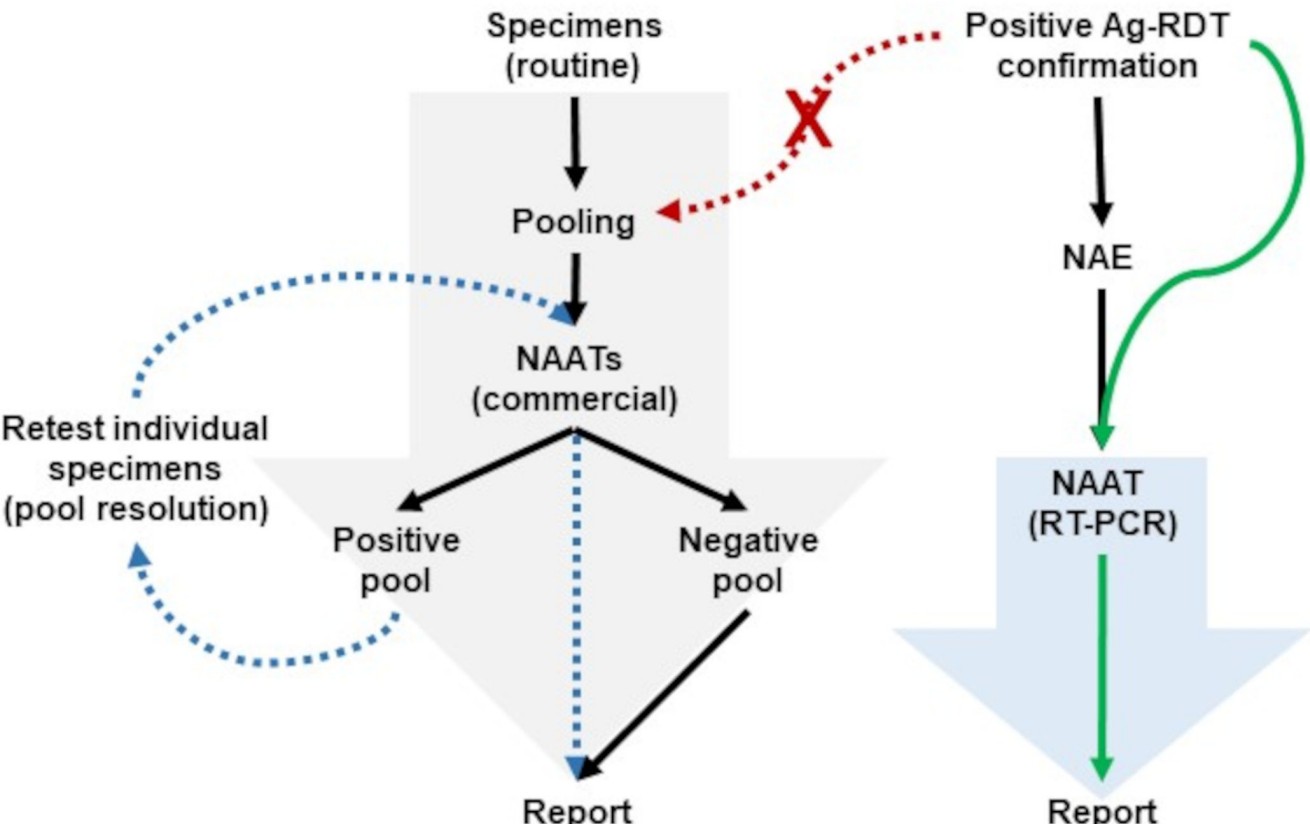

**FIG 1** Testing strategies used in Nova Scotia for low and high SARS-CoV-2 prevalence. The large gray and blue arrows represent the overall specimen throughput supported by commercial and laboratory-developed NAATs, respectively. (A) During low SARS-CoV-2 prevalence, nucleic acid extraction (NAE)-free RT-PCR (solid green arrow) supported positive Ag-RDTs confirmation without impacting routine specimen workflows using pooling on commercial NAATs. If positive Ag-RDT samples had been introduced into routine testing (red dashed arrows), it would have impacted testing strategies such as specimen pooling, where additional work would have been triggered to resolve positive pools (blue dashed arrows).

a misdiagnosis (prior to NAAT confirmation). This study compared the FPR and FDR in asymptomatic individuals.

To avoid false positive results and streamline NAAT-based confirmation of individuals testing positive by Ag-RDT, this study evaluated the performance of RT-PCR with and without NAE (i.e., Taqman PACMAN). Cost and workload were also modeled for the Taqman PACMAN compared to the same real-time RT-PCR paired with an NAE.

## MATERIALS AND METHODS

### Justifying NAAT confirmation for positive Ag-RDTs during periods of low SARS-CoV-2 prevalence

For Ag-RDT testing occurring within a week of symptom onset, the Abbott Panbio package insert described a sensitivity of 91.4% and specificity of 99.8% for the Ag-RDT nasopharyngeal (Np) swab collections when compared against Np swab RT-PCR. These values were considered the Ag-RDT performance best-case scenario. They were used to calculate the anticipated false discovery rate (FDR) based on prevalence using the following equations: FDR = [1 – positive predictive value (PPV)] and PPV = 100 × (sensitivity × prevalence)/[specificity × prevalence + ((1 – prevalence) × (1 – specificity))]. Similar calculations were performed using the WHO's minimum acceptable sensitivity of 80% and specificity of 97% for Ag-RDTs, and these criteria were considered the worst-case scenario (7). The sensitivity and specificity of the Panbio Ag-RDT for symptomatic

and asymptomatic SARS-CoV-2 cases were derived from a systematic review by Dinnes et al. (18), who reported sensitivity and specificity at 75.1% and 99.5% for symptomatic SARS-CoV-2 cases, and 48.9% and 98.1% for asymptomatic individuals. Data from the COVAG study by Wertenauer et al. (19) were also included who described sensitivity and specificity of 74.32% and 99.73% for symptomatic SARS-CoV-2 cases, and 23.28% and 99.96% for asymptomatic individuals.

## Ag-RDT asymptomatic testing

To understand real-world data on Ag-RDT performance in low prevalence settings, data were used from asymptomatic community testing sites coordinated by the Test-to-Protect volunteers and coaches or the Praxes Medical Group. All Ag-RDTs were performed and interpreted according to manufacturer instructions using the nasal or Np swab formulations of the Panbio COVID-19 Ag Rapid Test Device (Abbott Point of Care, Mississauga, ON, Canada) (8–11).

## Specimen collection for positive Ag-RDT confirmation using NAAT

In Nova Scotia, confirmatory NAAT testing of positive Ag-RDTs was initiated on 21 November 2020, with Ag-RDT deployment, at a time when community SARS-CoV-2 activity was low. Positive Ag-RDT confirmation was paused on 21 December 2021, as SARS-CoV-2 transmission had become widespread in the community (Fig. S1). Following positive Ag-RDT results in community testing centers, individuals were asked to return to testing sites for specimen recollection for NAAT-based confirmatory testing. Health providers collected Np swabs and specimens were transported to Nova Scotia hospital laboratories (3–5). With supply chain constraints during the first years of the pandemic, collection devices for NAAT-based testing varied. Np swab included nylon flocked swabs from Copan Diagnostics (Murietta, CA), Miraclean (Guangdong, China), or the AccuViral Collection Kit (AccuGene, San Diego, CA). Swabs were transported in universal transport media (UTM) (Copan Diagnostics, Murietta, CA) or viral transport medium (VTM) from Rodoxica (Little Rock, AR) or Yocon Biology Technology Company (Beijing, China).

## Diagnostic testing for SARS-CoV-2 using NAATs

Commercially available NAATs routinely used for diagnostic testing were carried out in Nova Scotia hospital laboratories according to the manufacturer's instructions on the Xpert Xpress SARS-CoV-2 assay (Cepheid, Sunnyvale, CA), the Aptima SARS-CoV-2 assay on the Panther system (Hologic, Inc., San Diego, USA), or the cobas SARS-CoV-2 test on the cobas 6800 instrument (Roche Diagnostics, Mannheim, Germany) (3–5). Following an NAE, a laboratory-developed test (LDT) was also used for SARS-CoV-2 diagnostic testing. In short, total nucleic acids (TNA) was extracted from 200 µL of the UTM or VTM fluid on a MagNA Pure 96 instrument (Roche Diagnostics Ltd.) according to the manufacturer's instructions. TNAs were eluted in a volume of 50 µL, and 5 µL was used as the template in the LDT reactions. The LDT was a real-time RT-PCR designed by the British Columbia Centre for Disease Control (BCCDC; Vancouver, BC) and targeted both SARS-CoV-2 envelope (E) and an RNA-dependent RNA polymerase (RdRp) (2). LDT results were interpreted based on cycle threshold (Ct) values. Results were considered positive if both target Ct values were <35 with exponential amplification. Results were considered indeterminate if there was exponential amplification and either target Ct value was between 35 and 37, or if single target detections were observed. Results were considered negative if both the E and RdRp Ct values were absent or if Ct values were >37.

## NAE-free real-time RT-PCR (Taqman PACMAN)

The NAE-free RT-PCR was identical to the LDT used for SARS-CoV-2 diagnostic testing with two exceptions. First, instead of NAE, 5 µL of specimen UTM (or VTM) was used as template in the RT-PCR reaction rather than the extracted TNAs. Second, the interpretation of Ct values was changed to reflect the 3 Ct value shift observed during the

method validation. Results for the NAE-free RT-PCR were in considered positive if both E and RdRp Ct values were <38 with exponential amplification. Results were considered indeterminate if either target Ct value fell between 38 and 40, or if single target detection were observed. Results were considered negative in the absence of Ct values or if Ct values were >40.

## Validation of the Taqman PACMAN

For validation of the Taqman PACMAN, the real-time RT-PCR was compared with and without NAE in three sets of experiments.

1. Limit of detection (LoD). Triplicate values from three independent 10-fold serial dilutions ($n = 9$) of a positive SARS-CoV-2 Np swab specimen were used to compare Ct values to estimate the limit of detection (LoD) of both methods. A similar series of twofold serial dilutions were performed in replicates ($n = 9$) for SARS-CoV-2 dilutions with results falling near the LoD. For quantification, Ct values of the RT-PCR with NAE were compared to a standard curve generated with quantified SARS-CoV-2 (2, 12) provided by the National Microbiology Laboratory (NML; Winnipeg, MB). The LoD of each method was estimated using a Probit analysis (2) at a probability of 95% confidence using MedCalc software version 22. Results were expressed as $\log_{10}$ copies/mL.

2. Comparison using samples tested with commercial SARS-CoV-2 NAATs. There were 752 clinical specimens (352 positive; 400 negative) characterized by commercial NAATs in hospital laboratories that were used to assess the performance of RT-PCR with and without NAE. Testing of both methods occurred in parallel within 24 h of testing with commercial NAATs. Specimens were stored at 4°C until testing was complete and then archived at −80°C.

3. Comparison using SARS-CoV-2 Ag-RDT positive samples. There were 344 specimens recollected from individuals testing positive by Ag-RDT at asymptomatic testing sites that were evaluated. Specimens were simultaneously processed using RT-PCR with and without NAE, as well as commercial NAATs. Specimen handling and storage were identical to those used with NAAT-positive samples.

## Method comparisons and statistical analyses

The RT-PCR with NAE was used as the reference method to assess the performance of the Taqman PACMAN. Discrepant results were resolved from consensus results (at least two of three) of commercial NAATs (i.e., Xpert, cobas, or Panther). Comparisons with these NAAT results were used to assign Taqman PACMAN or Ag-RDT results as true positive (TP), true negative (TN), false positive (FP), and false negative (FN). Descriptive statistics were performed using online software (https://www.omnicalculator.com/statistics/mcnemars-test). For comparison of the real-time RT-PCR with and without NAE, positive percent agreement (PPA) and negative percent agreement (NPA) were calculated from $2 \times 2$ contingency tables with 95% confidence intervals (CI) using the following equations: PPA = [TP/(TP + FN)] × 100 and NPA = [TN/(FP + TN)] × 100. A McNemar's test with Yates correction was used to assess differences, and $P \leq 0.05$ was considered statistically significant. Ag-RDT characteristics of interest included the % positivity (i.e., % Ag-RDT positive/total Ag-RDTs administered), FPR = [FP/TN + FP], and FDR = [FP/TP +FP].

## Impact of prevalence on the laboratory resources required to sustain confirmation of positive Ag-RDTs

Predictive cost estimation was performed to illustrate the impacts of SARS-CoV-2 prevalence on the laboratory resources needed to sustain confirmation of positive Ag-RDT results using RT-PCR with and without NAE (i.e., equipment, RT-PCR tests, human resources, and estimated costs). A population size of 100,000 administered Ag-RDTs was used as a surrogate of 1/10 of the total Nova Scotia population of 1,019,725 individuals.

With this population, the expected % positive Ag-RDTs generated at a prevalence of 5% would be 5,000, a value that would infringe on the maximum daily collection capacity of the province. The model only assumed laboratory costs and human resources including clerks and administrative support for entering patient information into the laboratory information system, medical laboratory assistants for labeling, aliquoting, and organizing specimens for testing, and medical laboratory technologists to perform the NAE (if needed), RT-PCR, data analysis, and reporting. The model only accounted for hand-on personnel time for tasks performed. The cost of the Ag-RDTs, specimen recollection, and other program support were not considered. Instruments at baseline used six thermocyclers available in the Division of Microbiology, Central Zone, Nova Scotia Health, as this was the location where NAE-free RT-PCR was evaluated and implemented. Additional instrumentation (i.e., extractors and thermocyclers) required at high prevalence was incorporated into the total costs, where appropriate.

## RESULTS

### Justifying the need for NAAT confirmation of Ag-RDTs positive results during low prevalence

High FDRs were prominent at low SARS-CoV-2 prevalence and negligible at high prevalence (Fig. 2). Despite sensitivity differences between the Abbott Panbio Ag-RDT package insert and those described in systematic reviews or other large studies for symptomatic or asymptomatic SARS-CoV-2 cases (18, 19), these had little impact on FDRs. However, even small differences in specificity in this study were demonstrated to greatly impact FDRs. In the COVAG study (19), only a small difference was observed between the Panbio Ag-RDT specificity for symptomatic and asymptomatic cases, and little differences were noticed for anticipated FDRs. With specificity values described for the same Ag-RDT in a systematic review (18), the FDR was equivalent to the kit insert for symptomatic cases, whereas for asymptomatic cases, the FDR overlapped with the trends seen with minimum acceptable criteria from the WHO (7). Therefore, the variability of FDR in multiple studies supports an investigation into the benefits of NAAT confirmation of Ag-RDT positive results during low prevalence, particularly for asymptomatic testing.

### Assessment of Ag-RDT performance in Nova Scotia

Ag-RDTs were first deployed in Nova Scotia on 21 November 2020 in community-based testing sites for asymptomatic individuals. Both Ag-RDT and NAAT confirmation results were captured until confirmatory testing was discontinued on 21 December 2021 (Fig. S1) (8–11). During this period, 331,364 Ag-RDTs were administered and the weekly Ag-RDT positivity varied from 0.00% to 0.47%, with an overall average of 0.25% (Fig. 3). Ag-RDT positivity remained low until late December when it rose to >6% with the rise of the SARS-CoV-2 omicron variant (Fig. 3A; Fig. S1), and confirmatory testing was discontinued (9, 10). Before this date, confirmatory NAAT results were documented on 88.0% (724/823) of the positive Ag-RDT results. The proportion NAAT-negative individuals with a false positive Ag-RDT (i.e., FPR) varied weekly from 0.0% to 100.0%.

Outside peaks of SARS-CoV-2 activity with pandemic waves, Ag-RDT positivity was low, and the FDR was highly variable from 0% to 100%. More precisely, from November 2020 to March 2021, Ag-RDT positivity varied markedly as did the %FDR. Some variations in %FDR can be explained by low numbers of positive results failing to confirm with NAAT where FDR reached 100%, as well as the absence of positive detections causing the FDR to reach 0%. Other trends were more revealing. From the study period of November 2020 to November 2021, peak Ag-RDT positivity and testing was observed in April 2021 and then declined progressively up to June 2021. Conversely, the %FDR increased during this period. A similar trend was seen from July to September 2021. In November 2021, there was a large increase in Ag-RDT positivity consistent with trends seen with NAAT results (Fig. S1) that was attributed to widespread SARS-CoV-2 activity

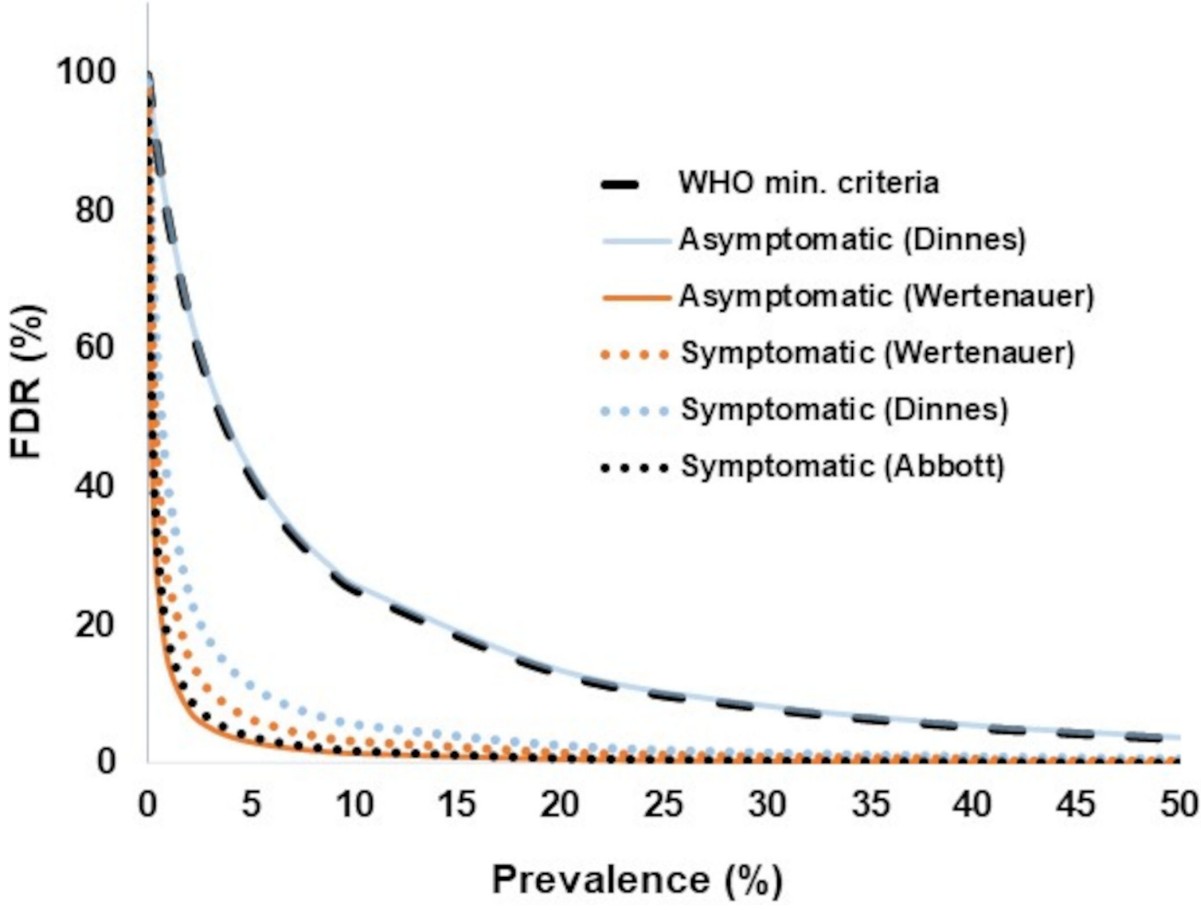

**FIG 2** Anticipated impact of prevalence on the Abbott Panbio Ag-RDT false discovery rate (FDR). FDRs were modeled in relation to prevalence based on sensitivity and specificity values described by (i) the manufacturer "best case scenario" at 91.4% and 99.8% (dotted black line), (ii) symptomatic SARS-CoV-2 cases in systematic review (18) at 75.1% and 99.5% (dotted blue lines), (iii) asymptomatic cases in the same review (18) at 48.9% and 98.1% (solid blue lines), (iv) symptomatic cases in the COVAG study (19) (dotted orange lines) at 74.32% and 99.73%, (v) asymptomatic cases in the COVAG study (19) at 23.28% and 99.96% (solid orange lines), and (vi) "worst case scenario" represented as the World Health Organization's (WHO) minimal criteria for Ag-RDTs test performance at 80% and 97% (dashed black line).

in the community with the rise of the omicron variant. During this time, the %FDR plummeted and remained low in December 2021 and in January 2022 (9, 10).

Overall, the FPR was low at 0.067% (222/330,763) in settings of low SARS-CoV-2 prevalence, whereas the %FDR was high at 30.7% (222/724). Unfortunately, calculating FPR and FDR for Ag-RDTs in other settings (e.g., workplaces, schools, households) was not possible given the lack of systematic capture of the overall testing numbers in Nova Scotia.

## Validation of NAE-free RT-PCR for Ag-RDT confirmation during low SARS-CoV-2 prevalence

Using triplicate values from three independent 2- and 10-fold serial dilutions of SARS-CoV-2 from a positive Np swab specimen, the NAE-free RT-PCR (i.e., Taqman PACMAN) was 10-fold less sensitive than the same RT-PCR with NAE, with LoDs of 3.286 $\log_{10}$ copies/mL and 2.356 $\log_{10}$ copies/mL, respectively. Compared to the RT-PCR with NAE, the NAE-free method showed an average Ct value shifts for E gene of 3.37 ± 0.96 and 3.24 ± 0.69 for RdRp (Fig. 4A and B).

Next, initial clinical assessment of the RT-PCR with and without extraction used 752 specimens tested using commercial NAATs. All 400 negative specimens were accurately

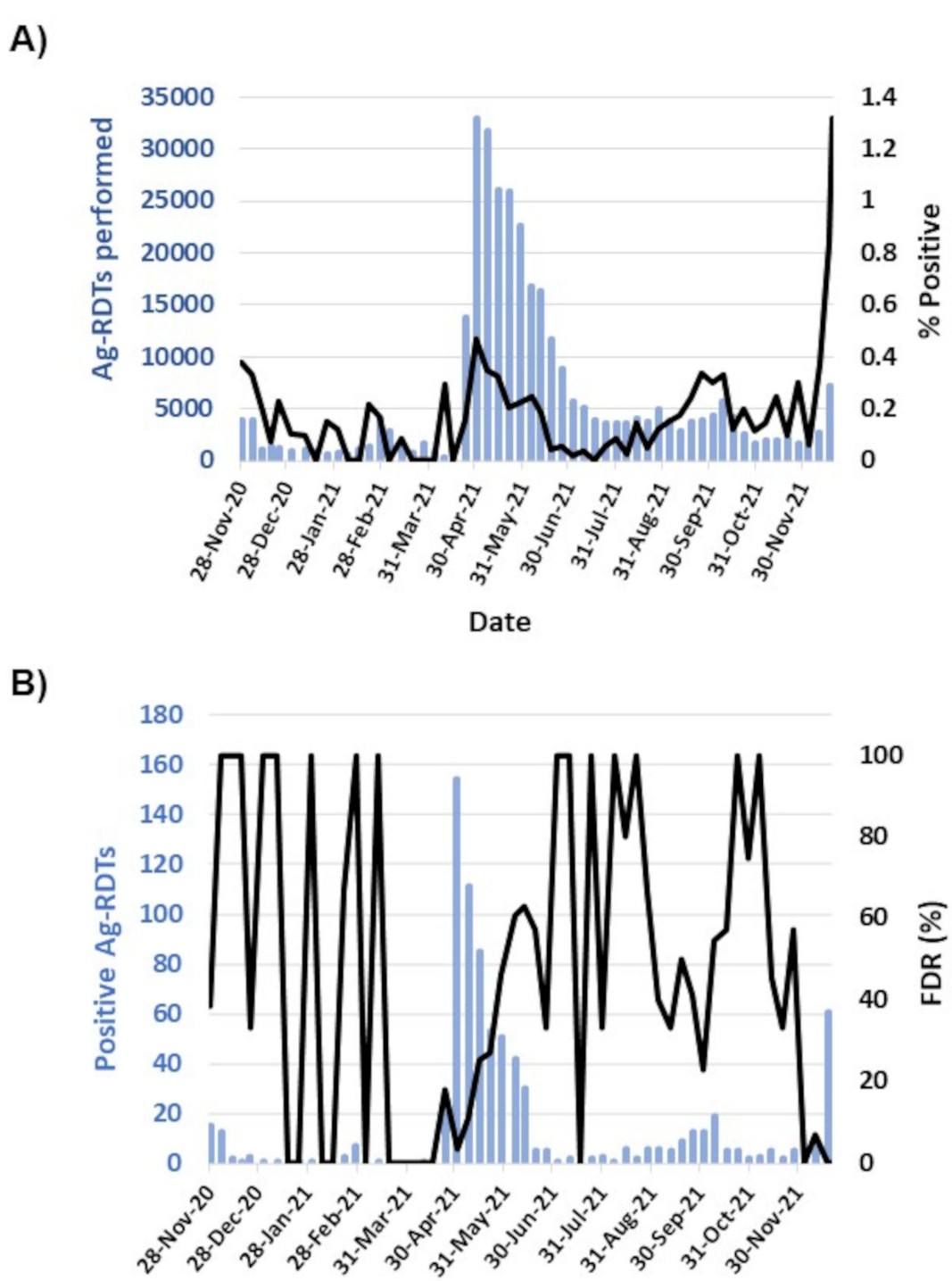

**FIG 3** Justifying positive Ag-RDTs confirmation with NAAT during periods of low SARS-CoV-2 prevalence. (A) SARS-CoV-2 Ag-RDT positivity and number of Ag-RDTs performed in asymptomatic community testing sites from 21 November 2020 to 21 December 2021. (B) Number of positive Ag-RDTs detected and proportion confirmed by NAAT following specimen recollection.

identified by NAE-free RT-PCR and RT-PCR with NAE, giving an NPA of 100.0% (95% confidence intervals: 99.1% to 100.0%). However, a statistically significant lower PPA at 96.0% (93.4% to 97.8%) was seen with NAE-free RT-PCR, detecting 338 of the 352 SARS-CoV-2 positive specimens (Fig. S2). The 14 discrepant results had E gene Ct values

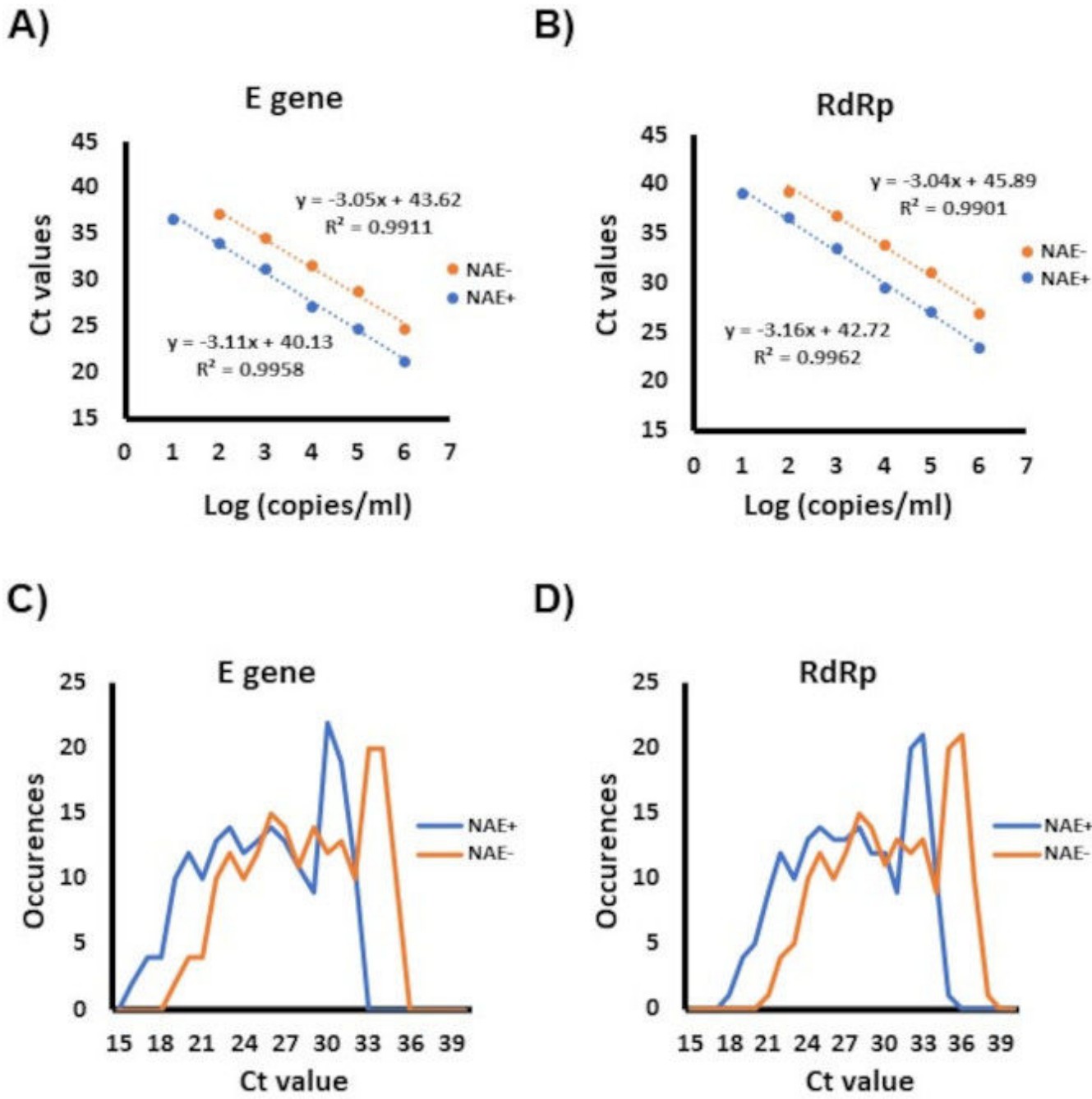

**FIG 4** Comparison of real-time RT-PCR with and without NAE. For analytical sensitivity comparisons, 10-fold serial dilutions of a quantified SARS-CoV-2 specimen were used to compare RT-PCR with (blue) and without NAE (orange). The average threshold cycle (Ct) values obtained for each dilution are displayed for both Rt-PCR target: (A) E gene and (B) RdRp. In (C) and (D) are the distribution of Ct values for RT-PCR with (blue) or without NAE (orange) obtained from 257 positive NAAT specimens recollected from individuals that tested positive with Ag-RDTs. C represents the results of the E gene target, whereas D represents those of RdRp.

> 35 (ranging from 35.27 to 36.95) or RdRp Ct values > 38 (ranging from 37.24 to 39.80) and would have been considered indeterminate with the RT-PCR with NAE (Table S1). The analytical sensitivity differences between the RT-PCR with and without NAE suggest that some specimens may potentially be missed with the NAE-free RT-PCR if used in any setting outside those of Ag-RDT assessed in this study.

During an internal laboratory audit of Ct values obtained from specimens obtained from Ag-RDT positive individuals, no RT-PCR Ct values were observed above 32 for E gene and 35 for RdRp (data not showed). Given this, it was hypothesized that the Taqman PACMAN NAE-free RT-PCR should be sufficiently sensitive to confirm positive Ag-RDTs and implementation of this method should streamline laboratory workflow (Fig. 1). To evaluate this hypothesis, 344 Np specimens collected from individuals testing positive by Ag-RDT at asymptomatic community test sites, all were concordant between the RT-PCR in the presence or absence of an NAE (Fig. S2). Compared to the RT-PCR with NAE, the NAE-free protocol showed ~3 Ct shifts consistent with those seen in the LoD experiments (Fig. 4C and D). Of note, 25.9% (87/344) of the specimens recollected from positive Ag-RDTs were negative by RT-PCR regardless of the presence or absence of an NAE, suggesting these were false positive Ag-RDTs. These 87 results were confirmed negative using three commercial NAATs (Xpert, cobas 6800, and Panther assays).

## Estimating the impact of Ag-RDT confirmation on laboratory resources

To estimate the impact of an NAE-sparing Ag-RDT confirmatory PCR testing and SARS-CoV-2 prevalence on Nova Scotia laboratory resource usage, cost, workload, and resources were calculated (Fig. 5). At a prevalence below 2%, a population of 100,000 would generate 2,000 positive Ag-RDTs, which would be manageable in a single work shift for NAE-free RT-PCR but would require three full work shifts and increased staffing using the same RT-PCR with NAE. As NAE-free RT-PCR results would be available more rapidly than the RT-PCR with NAE (approximately 90 min vs 3 h), NAE-free RT-PCR would require fewer RT-PCR runs (Fig. 5A), fewer work shifts and staffing (Fig. 5B), the PCR costs would be approximately fivefold less expensive with NAE-free RT-PCR, and the overall costs would be a fraction of those of RT-PCR with NAE (Fig. 5C). Little impact was observed with the NAE-free RT-PCR approach up to a prevalence of 4.5% other than a small proportional increase in RT-PCR costs and staffing required to meet the shift requirements. For NAE-free RT-PCR, the breaking point where this method would not be able to sustain Ag-RDT confirmation without additional human resources and instrumentation was at 5% SARS-CoV-2 prevalence, as the resulting 5,000 specimens requiring NAAT confirmation daily from individuals testing positive by Ag-RDTs would challenge both the collection capacity of the province as well as laboratory testing capacity. In contrast, the need for additional laboratory support for testing strategies based on RT-PCR with NAE would become apparent at a prevalence of 2%, with significant investments required, including the procurement of additional instrumentation, human resources, work shift expansions, a greater than 5-fold increase in RT-PCR costs, and total costs nearly 50-fold greater due to the need for additional instrumentation (extractors and thermocyclers). These additional resources were not available at the time.

## DISCUSSION

When public health and hospital laboratories were challenged and often overwhelmed by the high demand for NAATs, Ag-RDTs were a welcomed addition to the diagnostic armamentarium. With their ease of use, rapid results, and acceptable performance characteristics, Ag-RDTs were broadly used during the COVID-19 pandemic (1, 6–11, 18–20). As Nova Scotia was the first Canadian province to deploy Ag-RDTs as a public engagement tool for asymptomatic testing in community testing sites, data were collected on Ag-RDT and subsequent confirmatory NAAT results (8–11). In this setting, the Ag-RDTs FPR was low at 0.067%, but the FDR was high at 30.7%, supporting NAAT-based confirmation of positive Ag-RDTs. While the need for confirmatory testing in settings of low disease prevalence is not a novel concept to epidemiologists or diagnostic laboratories for SARS-CoV-2 (20, 21) or other microorganisms (22, 23), mass asymptomatic population testing for SARS-CoV-2 using Ag-RDTs during the COVID-19 pandemic provided an excellent opportunity to demonstrated how an NAE-free RT-PCR (i.e., the Taqman PACMAN) was a simple solution for positive Ag-RDT confirmation during

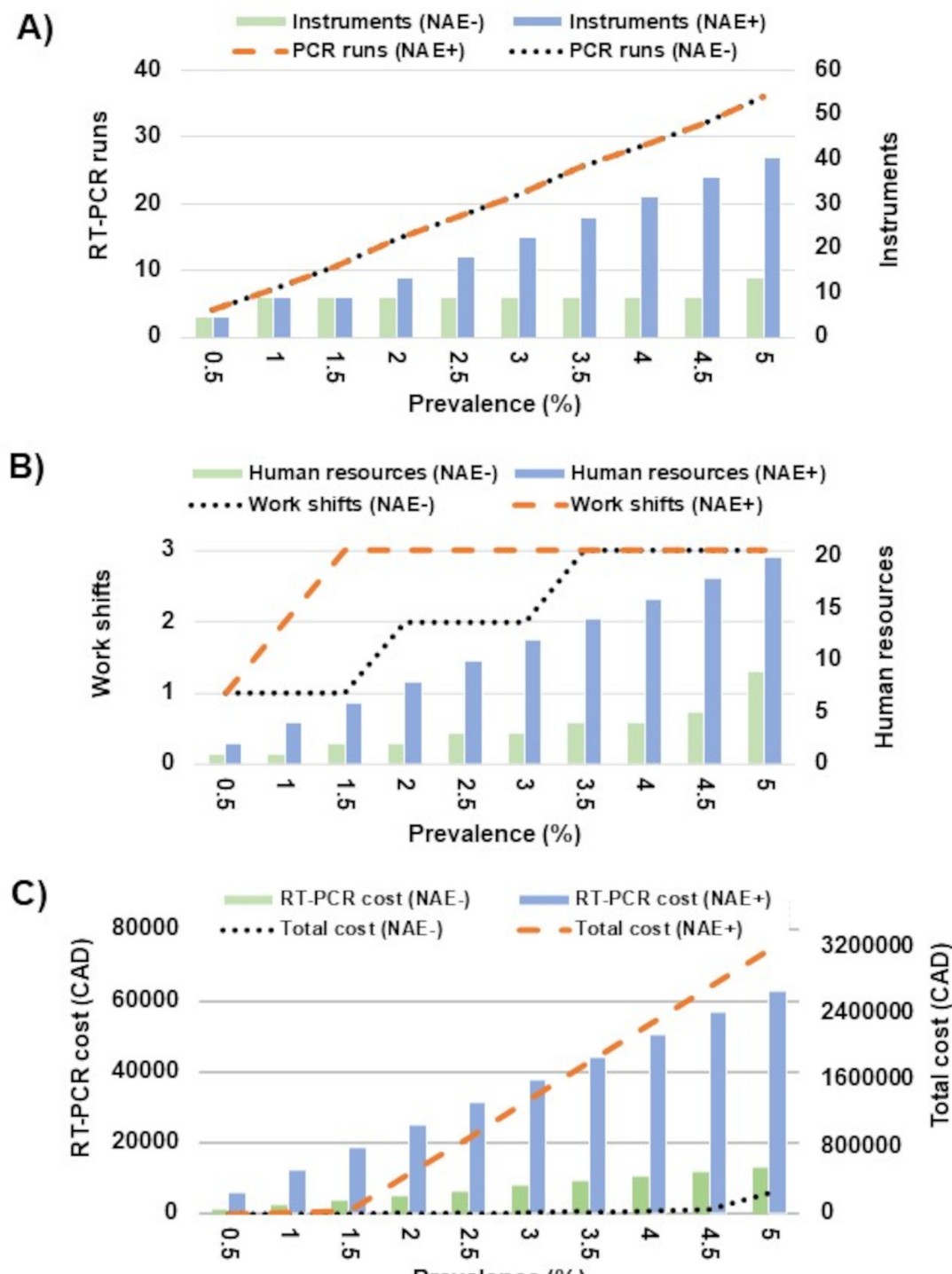

**FIG 5** Impact of SARS-CoV-2 prevalence on resources required to support Ag-RDT confirmation testing with RT-PCR with or without an NAE. Daily Ag-RDTs administered are depicted for a population of 100,000. The impact of SARS-CoV-2 prevalence is depicted: (A) for the number of RT-PCR runs and instruments required, (B) for the human resources and work shifts required, and (C) for RT-PCR and total costs.

periods of high testing volumes, low prevalence, and competing laboratory testing priorities.

It is well recognized that accurate SARS-CoV-2 diagnosis can aid the clinical management of cases and help interrupt transmission events through contact tracing and

isolation (1). Ag-RDTs are rapid and simple, allowing testing to be performed in virtually any setting (1). However, the sensitivity for SARS-CoV-2 Ag-RDTs is relatively lower compared to NAATs, but it has been argued that Ag-RDT positivity better reflects the period when SARS-CoV-2 is most communicable (24–26). To mitigate the impact of the relatively reduced sensitivity of Ag-RDTs compared to NAATs, frequent testing using Ag-RDTs was proposed (24–26); however, in a setting of low disease prevalence (e.g., home, school, workplace, or community-based testing), a higher frequency of testing also increases the risk of obtaining a false positive result, even if unintended. While the overall FPR was low at 0.07% (222/330,763) in asymptomatic testing sites, the broad use of Ag-RDTs in workplaces, schools, and other settings also increased the absolute number of false positive results. On an individual level, individuals testing positive for SARS-CoV-2 were legally required to self-isolate, and a false positive SARS-CoV-2 Ag-RDTs could result in absenteeism from school or work, personal financial loss if workplaces did not provide coverage for SARS-CoV-2 illness, delayed access to healthcare (e.g., elective surgeries), postponed travel and business opportunities, and psychological stress from the misdiagnosis, isolation, personal financial loss, or the fear of infecting others (21). On a population level, the potential impacts of false positive results include unnecessary public health investigations, business financial loss, misdirection of policies for closures (e.g., schools and workplaces), staffing shortages leading to difficulties in providing services (e.g., healthcare), missed opportunities for social gatherings including weddings and funerals, and other potential societal impacts arising from financial stress and isolation (21). With the potential for false positive Ag-RDTs during periods of low prevalence, NAAT confirmation was implemented since Ag-RDT deployment in Nova Scotia, which was consistent with national guidelines.

Implementing NAAT-based confirmatory testing for individuals with positive Ag-RDT was expected to be simple and feasible for diagnostic laboratories, as low specimen numbers were anticipated during periods of low disease prevalence. However, despite the low SARS-CoV-2 positivity during the alpha and delta variant waves compared to omicron (Fig. S1), there were sustained high levels of specimens submitted for NAAT testing to help inform testing strategies and ease public health restrictions. With the concomitant expansion of Ag-RDTs to broader settings, the number of individuals testing positive with Ag-RDTs requiring NAAT confirmation increased which impacted some of the testing strategies used in clinical laboratories (i.e., specimen pooling). In some settings of asymptomatic testing in Nova Scotia, point-of-care NAAT-based rapid diagnostic tests (NAAT-RDTs) were also used for positive Ag-RDTs confirmation (10, 27), which alleviated some burden on hospital laboratories. However, this strategy was not available or feasible for testing occurring in households, schools, and workplaces where individuals testing positive by Ag-RDTs were asked to book appointments for specimen recollection for NAAT testing occurring in hospital laboratories. For these specimens, the Taqman PACMAN NAE-free RT-PCR was a simple method to streamline Ag-RDT confirmation. Predictive estimation data demonstrate that without the Taqman PACMAN, confirmatory testing for positive Ag-RDTs would not have been possible for large populations without significant investments in instrumentation and human resources, which was not available at the time. While the benefits of NAE-free RT-PCR as a primary diagnostic test during mass SARS-CoV-2 testing have previously been recognized (15–17), the prevention of laboratory testing bottlenecks from supporting positive Ag-RDT confirmation following mass population testing had not previously been reported as an application of NAE-free RT-PCR.

While outside the scope of the study, it would be interesting to investigate whether certain individuals are more prone to false positive Ag-RDTs than others, and possible reasons. It would also be of value to investigate whether NAE-free RT-PCR could have other applications outside the confirmation of individuals testing positive by Ag-RDTs. Some laboratories have used NAE-free RT-PCR as a resource-sparing strategy or to sustain testing during periods when procurement of NAAT supplies was challenging (1, 15–17). For these and other applications, the NAE-free RT-PCR presented in this study

would be less sensitive than the same RT-PCR with NAE, and specimens containing lower SARS-CoV-2 viral loads might be missed. Regardless, for the proposed application of SARS-CoV-2 confirmation in individuals testing positive by Ag-RDTs, the sensitivity of NAE-free RT-PCR is sufficient.

This study was performed during low SARS-CoV-2 disease prevalence in asymptomatic community-based testing sites in a period where a single Ag-RDT kit was used (i.e., Abbott Panbio). The later expansion of Ag-RDTs into homes, schools, and workplaces used additional Ag-RDTs kits, but data for analyses of FPR and FDR were not available for these settings or Ag-RDT kits given the lack of systematic data capture in these settings. Another limitation of this investigation includes the inability to stratify FPR and FDR data by swab type and both Np and nasal swab sampling were available for Ag-RDTs. Nasal swabs were shown to be relatively less sensitive than Np swabs for the Ag-RDTs used in this study (11); however, specificity for these two specimen types was nearly identical and would not be expected to significantly skew the FDR or FPR. However, the accuracy of the FDR and FPR calculated in this study could be subject to some scrutiny. There were 99 positive Ag-RDTs in this study where NAAT results were not available. With this limitation, the FDR could range between 27.0% (222/823) and 39.0% (321/823) if none or all the 99 results were found to be false positive, respectively. In addition, recognizing Ag-RDT sensitivity could be as low as 48.9% for asymptomatic infections (18), applying this value to the 502 true positive Ag-RDTs observed in this study would increase the denominator to 1,027 and the resulting FDRs would be between 17.8% (222/1,249) and 23.8% (321/1,348). Therefore, in a worse case scenario, the true FDR for the Ag-RDT in asymptomatic testing sites would fall between 17.8% and 39.0%. As for the FPR, this study did not perform parallel testing of all Ag-RDTs administered with NAATs. If any Ag-RDT negative results would have been found to be NAAT positive, the true negative results used in this study would be overestimated and the FPR underestimated. In a low prevalence setting where many negative Ag-RDTs were observed, the contribution to the FPR of these false negatives is likely negligible. Overall, regardless of the uncertainty of the accuracy of the FDR and FPR reported in this study, their values are consistent with a previous Canadian paper (20) that used the same Ag-RDT (i.e., Abbott Panbio) during a similar timeframe (March to October 2021) to assess 903,408 asymptomatic individuals from 537 workplaces and reported a FPR of 0.05% and an FDR of 42%.

This study showed high FDRs in periods of low SARS-CoV-2 prevalence. The impact of high prevalence on FDR could not be assessed as positive Ag-RDT confirmation using NAAT was discontinued on 21 December 2021, after community SARS-CoV-2 activity became widespread (Fig. S1). Fortunately, the authors from this work had gathered Ag-RDT performance data on the same Ag-RDT kit to evaluate alternate swab collections during the rise of the omicron variant (9–11). Goodall et al. (9) reported no false positives Ag-RDT results during a 2-week period in January 2022, suggesting a 0.0% FDR in the asymptomatic population with an Ag-RDT positivity of 6.0% (31/520) using a combined nose/throat collection. In the following 2 weeks in the same setting, LeBlanc et al. (10) showed Ag-RDT positivity at 7.0% (364/5148), and with 10 false positives Ag-RDTs compared to NAAT-based testing, the FDR was 2.7% (10/364). Both these studies demonstrated low FDR in a setting of high SARS-CoV-2 prevalence, justifying the discontinuation of Ag-RDT confirmation in Nova Scotia during this period. However, it would be prudent to continue monitoring the FPR and FDR as disease prevalence evolves, particularly in the wake of pandemic waves.

The current study focused on specificity and false positives during a period of low weekly SARS-CoV-2 positivity (<0.47%) to demonstrate the need for NAAT-based confirmation of positive Ag-RDTs during periods of low prevalence. However, ongoing quality assurance should monitor the performance of Ag-RDTs or NAAT-based diagnostics tests through both periods of low and high prevalence. While outside the scope of this study, monitoring sensitivity and false negative rates is also important, particularly for RNA viruses like SARS-CoV-2 that are known to acquire mutations over time that may impact test performance characteristics (28–30). While it can be argued that

NAAT-based confirmation of positive Ag-RDTs is not required during periods of high disease prevalence, there could be value in re-assessing FPR and FDR as SARS-CoV-2 prevalence evolves, such as in the wake of pandemic waves or in settings not evaluated in the current study.

Overall, this work re-emphasizes the need for SARS-CoV-2 Ag-RDT confirmation during periods of low disease prevalence to avoid unintended consequences of a false positive result. With the rapid expansion of Ag-RDTs into broader communities, positive Ag-RDT confirmation would not have been possible in Nova Scotia without the streamlined NAAT testing provided by the Taqman PACMAN NAE-free RT-PCR. This method allowed rapid testing at nearly one-fifth of the cost of the NAE-associated RT-PCR and significantly reduced all resources required to support high-throughput testing, including human resources, instrument requirements, and overall costs. The Taqman PACMAN also streamlined NAAT-based confirmation of positive Ag-RDTs confirmation in a testing stream that was independent of routine processing, supporting testing demands in Nova Scotia during the SARS-CoV-2 alpha, delta, and beginning of the omicron variant waves. During a time of SARS-CoV-2 endemicity, these data support Taqman PACMAN as a rapidly deployable, financially efficient strategy that also conserves skilled workforce and laboratory resources in low disease prevalence to maximize infection awareness and minimize laboratory and societal disruption.

## ACKNOWLEDGMENTS

The authors would like to thank the National Microbiology Laboratory (NML) for providing quantified SARS-CoV-2 used for the LoD analyses. The authors would also like to thank the community participants and the Test-to-Protect (T2P) volunteers and coaches as well as members of the Praxes Medical Group who were instrumental in asymptomatic community test sites. The authors like to recognize the laboratory staff in Nova Scotia hospitals who were flexible in adapting to new testing strategies, and all the health care staff and volunteers at specimen collection sites who supported testing initiatives.

This work received no private or public funding, except the NAE and RT-PCR kits required for assay evaluation were provided in kind by the Division of Microbiology, Department of Pathology and Laboratory Medicine (DPLM), Nova Scotia Health (NSH).

All authors were involved with data acquisition, analysis, and interpretation. G.M. and J.L. performed the NAE-free RT-PCR validation and drafted the initial manuscript. All authors contributed to and agreed with the content of the final manuscript version.

## AUTHOR AFFILIATIONS

[1]Division of Microbiology, Department of Pathology and Laboratory Medicine, Nova Scotia Health, Halifax, Nova Scotia, Canada
[2]Department of Pathology, Dalhousie University, Halifax, Nova Scotia, Canada
[3]Department of Microbiology and Immunology, Dalhousie University, Halifax, Nova Scotia, Canada
[4]Department of Medicine, Dalhousie University, Halifax, Nova Scotia, Canada
[5]Nova Scotia Provincial Public Health Laboratory Network (PPHLN), Halifax, Nova Scotia, Canada
[6]Praxes Medical Group, Halifax, Nova Scotia, Canada

## AUTHOR ORCIDs

Glenn Patriquin http://orcid.org/0000-0001-8674-4358
Jason J. LeBlanc http://orcid.org/0000-0003-0593-0357

## AUTHOR CONTRIBUTIONS

Gregory R. McCracken, Conceptualization, Data curation, Formal analysis, Investigation, Methodology, Project administration, Validation, Visualization, Writing – original draft,

Writing – review and editing | Glenn Patriquin, Conceptualization, Methodology, Project administration, Supervision, Writing – original draft, Writing – review and editing | Todd F. Hatchette, Conceptualization, Methodology, Project administration, Supervision, Validation, Writing – original draft, Writing – review and editing | Ross J. Davidson, Conceptualization, Project administration, Supervision, Writing – original draft, Writing – review and editing | Barbara Goodall, Conceptualization, Data curation, Investigation, Methodology, Project administration, Supervision, Validation, Visualization, Writing – original draft, Writing – review and editing | Lisa Barrett, Conceptualization, Data curation, Investigation, Project administration, Supervision, Validation, Writing – original draft, Writing – review and editing | James MacDonald, Conceptualization, Investigation, Methodology, Project administration, Resources, Supervision, Validation, Writing – original draft, Writing – review and editing | Charles Heinstein, Conceptualization, Funding acquisition, Investigation, Methodology, Project administration, Supervision, Validation, Writing – original draft, Writing – review and editing | Janice Pettipas, Conceptualization, Data curation, Methodology, Project administration, Writing – original draft, Writing – review and editing | John Ross, Conceptualization, Data curation, Investigation, Project administration, Supervision, Validation, Writing – original draft, Writing – review and editing | Jason J. LeBlanc, Conceptualization, Data curation, Formal analysis, Investigation, Methodology, Project administration, Resources, Supervision, Validation, Visualization, Writing – original draft, Writing – review and editing

## ETHICS APPROVAL

This project was part of a quality initiative and was therefore exempt from review by the Nova Scotia Health Research Ethics Board (submission number 1027644). Specimens tested were obtained from consenting participants, and all data related were provided anonymized, deidentified, and used solely with the intent to evaluate the performance characteristics of testing programs used in Nova Scotia.

## ADDITIONAL FILES

The following material is available online.

### Supplemental Material

**Fig. S1 (Spectrum04073-23-s0001.pdf).** NAAT testing results in Nova Scotia spanning timelines relevant for this study.
**Fig. S2 (Spectrum04073-23-s0002.pdf).** Correlation between threshold cycle values obtained with real-time RT-PCR with and without nucleic acid extraction.
**Table S1 (Spectrum04073-23-s0003.pdf).** Summary of threshold cycle values from the 14 discrepant results.

### Open Peer Review

**PEER REVIEW HISTORY (review-history.pdf).** An accounting of the reviewer comments and feedback.

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
