## [Reviewer comments · Microbiology Spectrum]

Microbiology Spectrum

Taqman PACMAN: A simple molecular approach for positive rapid antigen test confirmation during periods of low prevalence

Gregory McCracken, Glenn Patriquin, Todd Hatchette, Ross Davidson, Barbara Goddall, Lisa Barrett, James McDonald, Charles Heinstejn, Janice Pettipas, John Ross, and Jason LeBlanc

Corresponding Author(s): Jason LeBlanc, Nova Scotia Health Authority

Review Timeline:

Submission Date:	December 11, 2023
Editorial Decision:	February 29, 2024
Revision Received:	March 4, 2024
Accepted:	March 13, 2024

Editor: Oliver Laeyendecker

Reviewer(s): The reviewers have opted to remain anonymous.

Transaction Report:

DOI: <https://doi.org/10.1128/spectrum.04073-23>

Re: Spectrum04073-23 (Taqman PACMAN: A simple molecular approach for positive rapid antigen test confirmation during periods of low prevalence)

Dear Dr. Jason J LeBlanc:

Thank you for the privilege of reviewing your work. Below you will find my comments, instructions from the Spectrum editorial office, and the reviewer comments.

Revision Guidelines

Sincerely,
Oliver Laeyendecker
Editor
Microbiology Spectrum

Reviewer #1 (Comments for the Author):

McCracken et al. describes in their manuscript a nucleic-extraction free protocol to perform confirmatory molecular (NAAT) testing on positive rapid antigen specimens for SARS-CoV-2. In low prevalence settings, it is recommended that positive antigen test results for SARS-CoV-2 need to be confirmed by a NAAT. However, the extra nucleic acid extraction step may hinder lab workflow or costs. This manuscripts describes the development of a new nucleic-extraction free protocol and showed that the

NPA and PPA are both 100% between NAAT with and without nucleic acid extraction. Based on modeling analysis, they also showed that laboratory metrics such as costs and work shifts may be positively impacted with a nucleic-acid extraction-free protocol if the disease prevalence was <5%.

Major comments:

1. Since the LDT includes both E and RdRp as gene targets for SARS-CoV-2, please explain why you only chose RdRp as a single target for your NAE-free assay?
2. The concepts behind FPR and FDR are confusing. Given that your gold standard is NAAT test (line 227), then aren't the FPR and FDR the same? If the PCR test is negative, then would the antigen test be a falsely detected as positive, so hence a false positive? Please clarify and include the clarification earlier in the manuscript. Line 221 is pretty far into the manuscript as the FDR acronym have been used in the earlier part of the manuscript already.
3. Figure 3, line 227. The varying rates of FDR from 0-100% seem broad. Is it because for some months, there were barely any positive Ag-RDTs (e.g. March) which would make the FDA 0% and then for some months with 100% FDR, there were not many positives? Perhaps there could be a table that shows the actual number breakdown of total tests since the percentages may it seem more alarming.
4. Please clarify if the 752 previously characterized specimens were tested with/without nucleic acid extraction in parallel. Using results from previous testing may present as a confounding variable since the specimen in the fridge/freezer can lose stability and cause a decrease in viral load anyway.
5. Given that this is a test development paper and you mentioned that the nucleic-acid extraction free method produced Ct values of +3, it is important to include data on the limit of detection of this test. In addition, it is normal practice per clinical lab standards to perform a limit of detection test whenever we bring on a newly lab-developed test. Please show the data in line 257. Also, was a true limit of detection test done? For example, could you spike known concentrations of virus into NP swabs and then perform the NAE-free PCR to see if what the limit of detection of this Taqman PACMAN. This would provide more quantitative information to the performance of this test, rather than just Ct values. Plus, you extended the Ct values cut-off for your NAE-free protocol. Showing that your PCR can still detect low quantities of virus would be convincing.
6. Line 262. Please confirm if the language is correct. The NAE-free protocol showed a reduction in Ct values of -3 (line 262) but then in line 246, the NAE-free protocol increased the Ct by +3. So in your asymptomatic patient set, the Ct values were lower?

Minor comments:

1. Line 103 refers to this method as "commonly referred". Is the Taqman PACMAN used for other applications? If so, please include advantages and limitations to this method published by other groups in the Discussion section.
2. Line 298, please correct demonstrated to 'demonstrate'

Reviewer #2 (Comments for the Author):

The manuscript describes a Taqman PACMAN-based approach for rapid antigen confirmation. It especially evaluates the cost-effectiveness of nucleic acid-free PCR tests in low-endemic settings. The writing is generally good and scientifically sound and could be of immense benefit during pandemics.

Authors should take note of the few comments below:

1. I would like the authors to explain the revised criteria for the interpretation of the NA-free PCR tests. If the difference between the NA-based PCR and NA-free PCR is about 10 folds, what minimum cycling threshold did they consider as positive? Is it 3 Cts below the lower limit of the NA-PCR?
2. The authors did not include information of the age variation of patients. Viral load for SARS-CoV-2 is known to vary for different age groups. Could this have had influence on the results?

Reviewer #1 (Comments for the Author):

McCracken et al. describes in their manuscript a nucleic-extraction free protocol to perform confirmatory molecular (NAAT) testing on positive rapid antigen specimens for SARS-CoV-2. In low prevalence settings, it is recommended that positive antigen test results for SARS-CoV-2 need to be confirmed by a NAAT. However, the extra nucleic acid extraction step may hinder lab workflow or costs. This manuscript describes the development of a new nucleic-extraction free protocol and showed that the NPA and PPA are both 100% between NAAT with and without nucleic acid extraction. Based on modeling analysis, they also showed that laboratory metrics such as costs and work shifts may be positively impacted with a nucleic- acid extraction-free protocol if the disease prevalence was <5%.

Major comments:

1. Since the LDT includes both E and RdRp as gene targets for SARS-CoV-2, please explain why you only chose RdRp as a single target for your NAE-free assay?
 - As stated in lines 152-153, the LDT used with and without nucleic acid extraction targeted both RdRp and the E gene of SARS-CoV-2; and only dual positive (E and RdRp) were considered positive and results were interpreted based on both targets. We have added information around the interpretation of both targets for both the RT-PCR with and without NAE and appreciate the opportunity to revise it. We have also added figures to demonstrate summary of Ct results of both targets that justifies the cutoffs in Figure 4C and D.
2. The concepts behind FPR and FDR are confusing. Given that your gold standard is NAAT test (line 227), then aren't the FPR and FDR the same? If the PCR test is negative, then would the antigen test be a falsely detected as positive, so hence a false positive? Please clarify and include the clarification earlier in the manuscript. Line 221 is pretty far into the manuscript as the FDR acronym have been used in the earlier part of the manuscript already.
 - The acronyms are defined in the introduction on line 103-106. "this study assessed the weekly Ag-RDT false positive rate (FPR; the proportion of individuals testing negative by NAAT who tested positive using Ag-RDT) and the weekly false discovery rate (FDR, the proportion of positive Ag-RDTs that did not confirm using NAATs)"
 - For clarity, we added two sentences to explain the difference between FPR and FDR, and the rationale why these are both important. "Both FPR and FDR have different merits and differ only in their denominator. The FPR represents how frequent false positives occur in patients without disease, which is dependent on the number of negative results identified. As such, the impact of false positive results could potentially be masked during periods of low prevalence with mass population testing of asymptomatic individuals, where the number of individuals testing negative is large. In contrast, the FDR is the number of false positive Ag-RDT results that occurred in the individuals testing Ag-RDT positive, therefore in those suspected to have SARS-CoV-2. As these individuals were asked to self-isolate in this setting, a positive Ag-RDT might have caused individual impacts like financial and psychological stress. Compared to FPR, the FDR better reflects the proportions of individuals subjected to undue stress from a misdiagnosis (prior to NAAT confirmation). This study compared the FPR and FDR in asymptomatic individuals."

3. Figure 3, line 227. The varying rates of FDR from 0-100% seem broad. Is it because for some months, there were barely any positive Ag-RDTs (e.g. March) which would make the FDA 0% and then for some months with 100% FDR, there were not many positives? Perhaps there could be a table that shows the actual number breakdown of total tests since the percentages may it seem more alarming.

- We completely agree with percentages being impacted by the number of tests performed as well as the positivity rate. That is why it was presented in Figure 3A prior to presenting %FDRs in Figure 3B. It should be noted that the text did explain this concept to some extent: “It should be noted that with rise of the delta and omicron SARS-CoV-2 variants in April and December 2021, respectively, the %FDR fell with increasing numbers of Ag-RDTs positives (Figure 3B), which mirrored trends in SARS-CoV-2 detection seen with NAATs (Figure S1). Outside peaks of SARS-CoV-2 activity with these pandemic waves, Ag-RDT positivity was low, and the FDR was highly variable from 0% to 100%.”
- However, an additional more descriptive paragraph was added following the initial results to help describe these findings: “More precisely, from November 2020 to March 2021, Ag-RDT positivity varied markedly as did the %FDR. Some variations in %FDR can be explained by low numbers of positive results failing to confirm with NAAT where FDR reached 100%, as well as absence of positives detections causing the FDR to reach 0%. However, there are periods that are more revealing. From November 2020 to November 2021, the peak Ag-RDT positivity and testing was observed in April 2021 then declined progressively up to June 2021 while conversely, the %FDR increased. A similar trend was seen from July to September 2021. In November 2021, there was a large increase in Ag-RDT positivity consistent with trends seen with NAAT results and widespread activity with rise of the SARS-CoV-2 omicron variant, and the %FDR plummeted and remain low in December 2021 and January 2022.^{9,10} Overall, the complex interplay between Ag-RDT positivity, testing numbers, and %FDR make monitoring FDRs challenging, but the high variability of %FDR during periods of low prevalence are evident, supporting NAAT-based confirmation.”

4. Please clarify if the 752 previously characterized specimens were tested with/without nucleic acid extraction in parallel. Using results from previous testing may present as a confounding variable since the specimen in the fridge/freezer can lose stability and cause a decrease in viral load anyway.

- Yes, the specimens were tested in parallel with the LDT with and without NAE, and the experiments used specimens previously characterized by commercial NAATs during routine clinical testing (within 24h). No freeze/thaw cycles occurred. We have added the following sentence to the methods to provide these details: “For testing using clinical specimens, the RT-PCR with and without NAE occurred in parallel within 24h of testing with commercial NAATs, and specimen were stored at 4°C until testing.”

5. Given that this is a test development paper and you mentioned that the nucleic-acid extraction free method produced Ct values of +3, it is important to include data on the limit of detection of this test. In addition, it is normal practice per clinical lab standards to perform a limit of detection test whenever we bring on a newly lab-developed test. Please show the data in line 257. Also, was a true limit of detection test done? For example, could you spike known concentrations of virus into NP swabs and then perform the NAE-free PCR to see if what the limit of detection of this Taqman PACMAN. This would provide more quantitative information to

the performance of this test, rather than just Ct values. Plus, you extended the Ct values cut-off for your NAE-free protocol. Showing that your PCR can still detect low quantities of virus would be convincing.

- The experiment presented in figure 4A and B show the NAE-free RT-PCR is less sensitive than RT-PCR with NAE using a limit of detection (LoD) analysis performed with a characterized SARS-CoV-2 positive sample (including stock concentration estimation using a standard curve generated with quantified material provided by the National Microbiology laboratory). A true LoD analysis was performed with 9 replicates near the LoD and a Probit analysis and value are now included in the results. Note, we did go further and tested 2-fold dilutions near the LoD but no additional detections were noted that were reproducible, and therefore are not presented in the figure. However, quantified values of the LoD (using a Probit analysis) have been added to the results for method comparison.

6. Line 262. Please confirm if the language is correct. The NAE-free protocol showed a reduction in Ct values of -3 (line 262) but then in line 246, the NAE-free protocol increased the Ct by +3. So in your asymptomatic patient set, the Ct values were lower?

- Values on line 262 were corrected to reflect the increase in Ct values with NAE-free RT-PCR (decrease in sensitivity) compared to the method using NAE (rather than the previously presented deviations).

Minor comments:

1. Line 103 refers to this method as "commonly referred". Is the Taqman PACMAN used for other applications? If so, please include advantages and limitations to this method published by other groups in the Discussion section.

- The method was referred to Taqman PACMAN in our lab. This wording "in our laboratory" has been added to the introduction. The method itself has been used in other applications such as resolving positive pools during clinical testing, but this is the subject of another manuscript. Other labs have used NAE-free RT-PCR as noted in the discussion "While the benefits and limitations of NAE-free RT-PCR as a primary diagnostic test during mass SARS-CoV-2 testing have previously been recognized¹⁵⁻¹⁷, the prevention of laboratory testing bottlenecks from supporting positive Ag-RDT confirmation following mass population testing had not previously been reported as an application of NAE-free RT-PCR." However, additional sentences have been added to the discussion, with cautions of possible reduced sensitivity for other applications "It would also be of value to investigate whether NAE-free RT-PCR could have other applications outside the confirmation of individuals testing positive by Ag-RDTs. Some laboratories have used NAE-free RT-PCR as a resource sparing strategy or to sustain testing during periods when procurement of NAAT supplies was challenging^{1,15-17}. For these and other applications, the NAE-free RT-PCR presented in this study would be less sensitive than the same RT-PCR with NAE, and specimens containing lower SARS-CoV-2 viral loads might be missed. Regardless, for the proposed application of SARS-CoV-2 confirmation in individuals testing positive by Ag-RDTs, the sensitivity of NAE-free RT-PCR is sufficient."

2. Line 298, please correct demonstrated to 'demonstrate'

- Corrected.

Reviewer #2 (Comments for the Author):

The manuscript describes a Taqman PACMAN-based approach for rapid antigen confirmation. It especially evaluates the cost-effectiveness of nucleic acid-free PCR tests in low-endemic settings. The writing is generally good and scientifically sound and could be of immense benefit during pandemics.

Authors should take note of the few comments below:

1. I would like the authors to explain the revised criteria for the interpretation of the NA-free PCR tests.

If the difference between the NA-based PCR and NA-free PCR is about 10 folds, what minimum cycling threshold did they consider as positive? Is it 3 Cts below the lower limit of the NA-PCR?

- The Ct cutoffs have been clarified in the methods to account for both E and RdRp targets, and do account for the 3 Ct shift seen with NAE-free RT-PCR seen with the LoD data. For clarity, we have added additional sections to figure 4 (C and D) to more clearly demonstrate the rationale of the Ct cutoff adjustments needed for with the observed Ct shifts.

2. The authors did not include information of the age variation of patients. Viral load for SARS-CoV-2 is known to vary for different age groups. Could this have had influence on the results?

- The age of the patients would have no impact on the results of this study as a direct comparison was undertaken to compare RT-PCR with and without NAE, with results spanning a wide range of Ct values. Variables possibly impacting viral load would be more applicable to false negatives, not confirmation of positives Ag-RDT (the application of this study). However, we did add a couple sentences in the discussion to generate thoughts on the possible causes of false positives "While outside the scope of the study, it would be interesting to investigate whether certain individuals are more prone to false positive Ag-RDTs than others, and possible reasons."

Re: Spectrum04073-23R1 (Taqman PACMAN: A simple molecular approach for positive rapid antigen test confirmation during periods of low prevalence)

Dear Dr. Jason J LeBlanc:

Your manuscript has been accepted, and I am forwarding it to the ASM production staff for publication. Your paper will first be checked to make sure all elements meet the technical requirements. ASM staff will contact you if anything needs to be revised before copyediting and production can begin. Otherwise, you will be notified when your proofs are ready to be viewed.

Sincerely,
Oliver Laeyendecker
Editor
Microbiology Spectrum